# Mitochondria Transfer in Brain Injury and Disease

**DOI:** 10.3390/cells11223603

**Published:** 2022-11-14

**Authors:** Lauren H. Fairley, Amandine Grimm, Anne Eckert

**Affiliations:** 1Transfaculty Research Platform Molecular and Cognitive Neuroscience, University of Basel, 4002 Basel, Switzerland; 2Neurobiology Laboratory for Brain Aging and Mental Health, Psychiatric University Clinics, 4002 Basel, Switzerland

**Keywords:** mitochondria, brain, neuron, glia

## Abstract

Intercellular mitochondria transfer is a novel form of cell signalling in which whole mitochondria are transferred between cells in order to enhance cellular functions or aid in the degradation of dysfunctional mitochondria. Recent studies have observed intercellular mitochondria transfer between glia and neurons in the brain, and mitochondrial transfer has emerged as a key neuroprotective mechanism in a range of neurological conditions. In particular, artificial mitochondria transfer has sparked widespread interest as a potential therapeutic strategy for brain disorders. In this review, we discuss the mechanisms and effects of intercellular mitochondria transfer in the brain. The role of mitochondrial transfer in neurological conditions, including neurodegenerative disease, brain injury, and neurodevelopmental disorders, is discussed as well as therapeutic strategies targeting mitochondria transfer in the brain.

## 1. Introduction

Mitochondrial dysfunction has been observed in a wide range of brain states and pathologies, including normal brain aging, brain injury, and disease [1]. Often depicted as the powerhouses of the cell, mitochondria are not only the main generators of energy in the form of adenosine triphosphate (ATP) via oxidative phosphorylation (OXPHOS) and glycolysis, but they are also involved in other processes such as calcium homeostasis and apoptosis [2,3]. The term mitochondrial dysfunction encompasses a range of mitochondrial deficits, ranging from bioenergetic impairment, oxidative stress, mitophagy dysfunction, and altered mitochondrial dynamics. As one of the most highly energy-demanding organs in the body [4], the brain is particularly susceptible to mitochondrial impairments, and research has shown that mitochondrial dysfunction plays a crucial role in the pathogenesis of brain disorders [1,5,6,7,8]. As a result, a wide range of therapeutics targeting mitochondrial dysfunction in the brain have been tested, including pharmacologic approaches (neuroactive steroids, antioxidants, phenylpropanoids) [9,10,11,12,13,14] and lifestyle interventions (diet, exercise) [3,15,16]. However, the multifaceted and sometimes opposing modes of mitochondrial dysfunction in disease pathology render it difficult to target mitochondrial dysfunction as a whole.

Recently, a novel form of cell-to-cell signalling has been identified in the form of intercellular mitochondrial transfer. This process involves the active transfer of whole mitochondria out of the donor cells via nanotubes, extracellular vesicles, or other methods, where they are internalised by recipient cells and either incorporated into the mitochondrial network or further processed for degradation [17]. This ability to exchange mitochondria has been shown to exert a range of beneficial effects on mitochondrial function in recipient cells, such as increasing ATP levels, restoring mitochondrial membrane potential, and normalizing neuronal calcium dynamics [18,19,20].

Glia, consisting of astrocytes, microglia, and oligodendrocytes, perform a wide range of specialist functions in the brain, including removing pathogens and promoting neurorecovery following injury or disease. In addition, glia demonstrate a number of unique properties that make them uniquely positioned to participate in intercellular mitochondrial transfer in the brain. For instance, glia possess the capability to react rapidly to changes in neuronal function, whether by modulating physical contact, regulating neurotransmission, participating in synaptic pruning, and providing metabolic support [21,22]. Further, glia uniquely respond to inflammatory events in the brain by undergoing immune activation and mitochondrial metabolic reprogramming [23], a process which has been shown to increase intercellular mitochondrial transfer [24,25,26]. Intriguingly, glia appear to be more resistant to injury and stress-induced insults in the brain compared to neurons. For instance, whereas treatment with the chemotherapeutic drug cisplatin reduced neuronal survival and impaired mitochondrial function in neurons, no effect on these output measures was observed in astrocytes treated with equivalent doses [18]. Furthermore, astrocytes are more resistant to ischemic injury *in vitro* [27,28] and show increased survival in animal models of stroke [29]. At the molecular level, DNA-seq data demonstrated that astrocytes have higher concentrations of mitochondrial polymerase gamma (polγ) compared to neurons, which is critically involved in mitochondrial replication, mutagenesis, and repair of mitochondrial DNA [18,30]. Taken together, these findings suggest that glia exhibit greater resilience to injury and stress compared to neurons, and their ability to maintain a healthy pool of mitochondria may underlie this ability.

An increasing body of literature suggests that glia play a critical role in modulating cell viability in the brain via intercellular mitochondrial transfer. Specifically, glial–neuronal mitochondrial transfer has been shown to confer neuroprotective effects by enhancing neuronal viability following exposure to harmful stimuli, enhancing degradation of damaged or dysfunctional mitochondria, and modulating glia-mediated neuroinflammation. Thus, strategies aimed at enhancing intercellular mitochondrial transfer have been investigated as potential therapeutic targets for neurological disease. As the brain is one of the most studied organs in terms of mitochondrial impairments, this review focuses on new recent findings regarding glial–neuronal intercellular mitochondrial transfer in the brain. The role of mitochondrial transfer in disease and therapeutic strategies targeting mitochondrial transfer in the brain are discussed in detail.

## 2. Evidence of Intercellular Mitochondrial Transfer in the Brain

In the brain, bidirectional intercellular mitochondria transfer has been observed between glia and neurons, and also between glia and other glial cells both *in vitro* and *in vivo*. *In vitro*, astrocytes have been shown to transfer mitochondria to neurons in primary rat cortical astrocytes and neurons [18,20]. Similarly, Gao and colleagues [31] showed that astrocytes dynamically transfer mitochondria to neurons in neural cell lines, primary neural cells, and human pluripotent stem cell (hPSC)-derived neural cells. *In vivo* studies using artificial mitochondrial transplantation have also shown that astrocyte-derived mitochondria directly injected into the peri-infarct cortex of mice were present in neurons after 24 h [20]. Conversely, neurons have been shown to transfer damaged mitochondria to adjacent astrocytes in mouse primary neurons and astrocytes [31,32,33]. Intercellular mitochondrial transfer has also been observed between glia and other glia cells. *In vitro*, astrocytes are able to transfer mitochondria to other astrocytes in astrocyte cell lines, primary astrocytes, and hPSC-derived astrocytes [31] as well as microglia [34]. Similarly, microglia have been shown to transfer mitochondria to astrocytes in microglia cell lines and primary microglia [35]. Mitochondria transfer has also been observed between tumour-to-tumour cells and tumour-to-tumour microenvironment in the brain. For instance, mitochondria transfer has been observed between glioblastoma stem-like cells *in vitro* [36] as well as between glioblastoma cells and surrounding non-tumour astrocytes [37]. Fewer studies have investigated mitochondrial transfer in oligodendrocytes in the brain; however, Zhao and colleagues [34] reported that oligodendrocytes exhibit reduced capacity to internalize exogenous mitochondria, with less than 10% of oligodendrocytes exhibiting internalized mitochondria. Thus, they are not discussed in detail in this review.

## 3. Structural Mechanisms of Mitochondrial Transfer

Glial–neuronal mitochondrial transfer is mediated via a number of active processes including the release of extracellular vesicles, the formation of tunnelling nanotubes, and potentially other mechanisms (Figure 1, Table 1). These processes are discussed in detail below:

### 3.1. Extracellular Vesicles

Extracellular vesicles (EVs) are bilayer membranous structures that are secreted into the extracellular space. There are two main types of EVs: exosomes and microvesicles. Microvesicles are distinguished from exosomes based on their larger size (typically 0.1–1 μm in diameter), whereas exosomes are much smaller (30–150 nm in diameter) [38]. Astrocytes have been shown to shed EVs containing functional mitochondria ranging from 300 nm up to 8 μm in diameter [20,39]. Similarly, microglia release EVs containing mitochondria into the extracellular space, where they are internalised by astrocytes [35]. It should be noted that EVs containing mitochondria are differentiated from previously identified EV subtypes such as mitovesicles [40] and mitochondria-derived vesicles (MDVs) [41], which are small vesicles that transport mitochondrial-derived cargo but lack several mitochondrial structures such as cristae, mitochondrial ribosomes, and proteins found in mitochondria, such as Tomm20 [40].

### 3.2. Tunneling Nanotubes

Tunnelling nanotubes (TNTs) were first described in 2004 by Rustom and colleagues [42], who identified membranous channels comprised of F-actin that connect two or more cells and are involved in cell-to-cell communication [43]. TNTs were shown to transfer different organelles, including mitochondria, from one cell to another but also other cargo, including proteins (e.g., α-synuclein) and nucleotides [4,5,6,7,8,9,10,11,12,13,14,15,16,17,18,19,20,21,22,23,24,25,26,27,28,29,30,31,32,33,34,35,36,37,38,39,40,41,42,43,44]. Intercellular mitochondrial transfer via TNTs has been observed in astrocytes, microglia, and neurons. For instance, microglia have been shown to form a network of F-actin-positive intercellular membrane projections with neighbouring microglia that contain mitochondria [44]. TNT formation has also been observed between astrocytes and adjacent neurons in rat primary astrocyte and neuronal cells [33,45,46,47]. The direction of mitochondrial transfer between cells via TNTs is not fully understood; however, there is evidence to suggest that TNTs are formed in response to stress. For instance, exposure to hydrogen peroxide (H_2_O_2_) increased TNT formation between astrocytes *in vitro* [47]. Further, Wang and colleagues [45] reported that in astrocyte–neuron co-culture, the cells exposed to stressful stimuli, such as H_2_O_2_ or serum deprivation, develop TNTs towards the unstressed cells but not vice versa. Other studies have identified the small calcium-binding protein S100A4 and its putative receptor RAGE (receptor for advanced glycation end product) as guidance molecules that mediate the growth and direction of TNTs Wang [45]. However, further studies on the mechanistic pathway regulating TNT formation and directionality are needed. Table 1 summarizes findings of studies investigating the effects of mitochondria transfer via TNTs in different brain disease models and highlight the positive effect of this transfer [19,33,36,44,45,48,49,50].

Of note, Sartori-Rupp and colleagues found that mitochondria can be transported between neuronal cells via individual TNTs composed of a single, continuous bundle of parallel actin filaments [51]. This suggests that mitochondria directly bind to actin filaments and are transported via a microtubule-independent mechanism that remains to identified. 

### 3.3. Other Mechanisms

Although most evidence to date suggests that glial–neuronal mitochondrial transfer occurs via EVs or TNTs, a number of other intercellular transfer mechanisms have been proposed, including cell fusion and gap junctions [52]. Gap junctions are plasma membrane channels composed of connexins that have been shown to transfer small molecules [53], and more recently, whole mitochondria [54] between cells. Li and colleagues [55] observed mitochondrial transfer from bone marrow mesenchymal stem cells to motor neurons via gap junctions, and gap junction connexin32 is expressed in neurons. However, whether glia cells are capable of transferring mitochondria via gap junction has not yet been investigated. Cell fusion involves the direct fusion of two cellular membranes, which would theoretically allow for the transfer of cytoplasmic molecules, including mitochondria. However no direct evidence of cell fusion mediated mitochondrial transfer in glia or neurons has been observed. Further studies are needed to investigate whether glial–neuronal mitochondrial transfer can occur via gap junctions and cell fusion.

**Table 1 cells-11-03603-t001:** Mechanisms of Mitochondria Transfer in the Brain.

Method of Transfer	Cell Type		Disease Model/Stressor	Effects of Mito Transfer		Ref.
	Donor	Recipient		Donor	Recipient	
EVs	Neural stem cells	BMDM	LPS	N/A	-Increased mitochondrial fusion	[56]
					-Increased cellular respiration	
					-Reduced inflammatory gene profiles	
EVs	Primary human	N/A	ATP released from neighbouring cells	N/A	N/A	[39]
	Astrocytes					
EVs	Primary rat astrocytes	Primary rat neurons	Oxygen-glucose deprivation	N/A	-Increase ATP levels	[20]
					-Increased cell viability	
TNTs	PC12 cells	PC12 cells	UV light	N/A	-Decreased apoptosis	[48]
TNTs	MMSC	Primary astrocytes	Oxygen-glucose deprivation	-Increased transfer	-Restored bioenergetics	[49]
		and			-Increased proliferation	
		PC12 cells				
TNTs	MSC	Neural stem cells	Cisplatin	N/A	-Decreased apoptosis	[50]
					-Increased MMP	
TNTs	Primary mouse astrocytes	Primary mouse neurons	Compressed nitrogen–oxygen mixed gas	N/A	-Increased dendrite length	[19]
					-Increased transcription of mitochondrial synthesis-related genes	
TNTs	Primary mouse microglia	Primary mouse microglia	α-syn	N/A	-Decreased ROS levels	[44]
					-Decreased apoptotic signalling	
TNTs	Primary rat astrocytes	Primary rat astrocytes and neurons	H_2_O_2_ or serum depletion	N/A		[45]
TNTs	primary mouse neurons	primary mouse astrocytes	5xFAD	N/A	-Increased transmitophagy	[33]
TNTs	Glioblastoma stem-like cells	Glioblastoma stem-like cells	Irradiation	C1: no effect	N/A	[36]
				C2: increased transfer		

Abbreviations: EV, extracellular vesicle; TNT, tunnelling nanotube; BMDM, bone-marrow-derived macrophage; N/A, not applicable.

## 4. Effects of Mitochondrial Transfer in the Brain

Intercellular transfer of mitochondria between glia and neurons has been shown to serve three primary functions in the brain (Figure 2):To enhance cell viability via transferring healthy mitochondria to stressed/injured cells;To enhance degradation of dysfunctional mitochondria via transferring unhealthy mitochondria to healthy cells;To modulate glia-mediated neuroinflammation.

### 4.1. Enhancement of Cell Viability

Astrocytic transfer of mitochondria to neurons has been shown to exert a range of neuroprotective effects. For instance, mitochondrial transfer from astrocytes to neurons has been shown to increase neuronal survival, restore neuronal mitochondrial membrane potential, increase ATP levels, normalize neuronal calcium dynamics, and increase dendrite length *in vitro* [18,19,20]. Furthermore, inhibition of astrocytic mitochondria transfer increases neuronal vulnerability to cell death [57,58], suggesting that intercellular transfer of mitochondria between astrocytes and neurons plays a critical role in mediating neuroprotection.

Neuronal release of mitochondria has also been proposed as a “Help-Me” signal that promotes the transfer of healthy mitochondria from astrocytes under stressful conditions. For instance, increased release of defective mitochondria from neurons is observed *in vitro* following challenges such as acidosis, hydrogen peroxide (H_2_O_2_), N-methyl-D-aspartate (NMDA), or glutamate exposure [59]. Upon release into the extracellular space, these defective mitochondria are taken up by astrocytes, which in turn causes increased astrocytic expression of mitochondrial Rho-GTPase 1 (Miro-1), which is known to facilitate the transfer of healthy mitochondria from astrocytes to neurons [18]. Taken together, these findings suggest that astrocytes dynamically transfer mitochondria to neurons to enhance neuronal viability and confer neuroprotection following exposure to stressful stimuli.

### 4.2. Enhancement of Mitochondrial Degradation

Conversely, neurons have been shown to transfer damaged mitochondria to adjacent astrocytes, where they are degraded via a process termed “transmitophagy” [31,33,60]. This process has been shown to result in elevated mitochondrial membrane potential in recipient astrocytes, suggesting it may also enhance recipient cell mitochondrial function [31]. Intriguingly, Davis and colleagues [32] reported that the majority of mitochondrial degradation in retinal ganglion cells is performed by adjacent astrocytes, with a much smaller proportion being performed inside the neuronal cell body. In addition, Rhes protein, a critical regulator of mitophagy in the brain, has been shown to transfer between striatal neuronal cells via TNTs, where it binds to damaged mitochondria in the recipient cell, suggesting that neurons may also transfer mitophagy-enhancing proteins to aid the transmitophagy process [61]. 

Mitochondria have also been proposed to act as carriers of deleterious cargo to improve intercellular pathogenic clearance. For instance, microglia exposed to α-synuclein (α-syn), a neuronal protein found in Lewy bodies in synucleinopathies, including Parkinson’s disease (PD), have been shown to transfer both mitochondria and α-syn to neighbouring healthy microglia via TNTs, where the α-syn cargo is effectively degraded [44]. Taken together, these findings highlight an important role for intercellular mitochondrial transfer in aiding in the degradation of harmful stimuli and damaged mitochondria.

### 4.3. Modulation of Glia-Mediated Neuroinflammation

Intercellular mitochondrial transfer between glial cells has also been implicated in the regulation of inflammatory glial phenotypes in the brain. Astrocyte-released mitochondria appear to exert anti-inflammatory effects on microglia *in vitro*, as uptake of astrocytic mitochondria increased humanin levels in microglia, which was linked to increased levels of peroxisome proliferator-activated receptor-γ (PPARγ) and its transcriptional target manganese superoxide dismutase (Mn-SOD), both of which promote anti-inflammatory reparative microglial phenotypes [34]. This same study showed that astrocytic mitochondrial transfer also increased microglial phagocytosis of red blood cells. Mechanistically, the transfer of healthy mitochondria to immune cells has been proposed to modulate anti-inflammatory and phagocytosis-enhancing effects under inflammatory conditions via the promotion of OXPHOS. For instance, conditioned media from mesenchymal stem cells (MSCs) promoted anti-inflammatory phenotypes and enhanced phagocytic function in monocyte-derived macrophages treated with lipopolysaccharides (LPS) [26]. However, inhibition of OXPHOS using the ATP synthase inhibitor oligomycin completely prevented these anti-inflammatory effects and blocked phagocytic function. These findings suggest that intercellular mitochondria transfer may play a key role in regulating immunometabolic signalling the brain via enhancing OXPHOS and its associated immune functions.

Similarly, microglia are able to transfer mitochondria to astrocytes in microglia cell lines and primary microglia [35]. However, unlike astrocytic mitochondria transfer, activated microglia were shown to propagate inflammatory signals to astrocytes via the transfer of dysfunctional mitochondria, which triggered a proinflammatory A1 activation state in recipient astrocytes [35]. These microglia-activated astrocytes in turn released fragmented mitochondria into the extracellular space that triggered neuronal damage by lowering ATP production and mitochondrial membrane potential. However, whether microglia directly transfer deleterious fragmented mitochondria to neurons or whether they simply act as an intermediary between astrocytes and neurons remains unknown. 

Exposure to inflammatory stimuli has been shown to alter intercellular mitochondrial transfer in various ways. For instance, several studies have reported that exposure to the inflammatory stimuli LPS induces mitochondrial transfer in the lung [24] and between bone marrow stromal cells and hematopoietic stem cells *in vitro* [62]. In the brain, astrocytic transfer of mitochondria to neurons increases following exposure to stressful stimuli such as the chemotherapy drug cisplatin [18], and stimulation with LPS and nigericin increased the total amount of mitochondrial protein content present in the media of cultured primary microglia [35]. Conversely, inhibition of inflammation reduced the number of EVs released from microglia, but whether these EVs contained mitochondria is not known [63]. A recent study suggested that inflammatory activation may not alter the amount of extracellular mitochondria released from glia but rather the ratio of functional to dysfunctional mitochondria [35]. Stimulation with LPS and nigericin increased the total amount of mitochondrial protein content present in the media of cultured primary microglia but lowered the amount of functional mitochondria (defined by preserved mitochondrial membrane potential and ability to generate ATP) [35]. Similarly, astrocytes activated by a mixture of proinflammatory cytokines tumour necrosis factor α (TNF-α), interleukin 1α (IL1α), and complement component 1q (C1q) lowered the number of functional mitochondria released into the extracellular media [35]. 

Taken together, these findings suggest that exposure to inflammatory stimuli regulates intercellular mitochondrial transfer in the brain in a number of ways, including modulating the number and/or quality of transferred mitochondria, mediating the propagation of inflammatory signals between astrocytes and microglia, and regulating immunometabolic crosstalk.

### 4.4. Deleterious Effects

Additionally, deleterious side effects of intercellular mitochondrial transfer have been suggested in specific disease models. For instance, mitochondria have been hypothesized to serve as carriers of α-syn [64] in synucleinopathies, including Parkinson’s disease (PD), dementia with Lewy bodies (DLB), and multiple system atrophy (MSA). Valdinocci and colleagues reported that α-syn interacts with the mitochondrial outer membrane and that mitochondria-bound α-syn is observed in tunnelling nanotubes in neurons [65]. Furthermore, both amyloid-β and tau, two pathological hallmarks of Alzheimer’s disease (AD), are known to bind to mitochondria [66,67,68], and have been shown to propagate between cells via TNTs [69,70]. These findings also have implications for other disease-related cargo that are known to bind to mitochondria, such as SARS-CoV-2 [71], and suggest that under specific disease conditions, mitochondria may be hijacked as a means to facilitate cell-to-cell transfer of pathogens in the brain. However, direct evidence for intercellular mitochondrial transfer as a means of spreading disease-associated proteinopathy in the brain is still lacking.

## 5. Mitochondrial Transfer in Brain Injury and Disease

Mitochondrial dysfunction has been identified as a hallmark feature of brain injury and disease. Specifically, alterations in mitochondrial membrane potential, oxidative stress, ATP production, and mitochondrial dynamics are thought to play a causative role in the pathogenesis of neurodegenerative disorders, including Alzheimer’s disease [7,23], Parkinson’s disease [72], and multiple sclerosis [73], as well as stroke [74], traumatic brain injury [75], and neurodevelopmental disorders [76,77]. Recent evidence from animal models suggests that brain injury or disease induces mitochondrial transfer from glia to neurons and is associated with improved neurological outcomes. As a result, the mechanistic role and therapeutic potential of mitochondrial transfer in the brain has sparked widespread interest.

### 5.1. Brain Injury

Stroke is associated with a range of mitochondrial impairments in damaged brain regions, and maintaining mitochondrial integrity is crucial in promoting neuronal survival after ischaemic injury [74]. As a result, intercellular mitochondrial transfer has been identified as a promising therapeutic strategy for the treatment of stroke. Neuroprotective effects of mitochondrial transfer in stroke were first reported by Huang and colleagues, who demonstrated that local intracerebral or systemic intra-arterial injection of isolated hamster mitochondria into brain-ischemic rats significantly reduced neuronal death and restored motor performance [78]. Similarly, Hayakawa and colleagues showed that an influx of astrocytic mitochondria into adjacent neurons is observed following induction of focal cerebral ischaemia in mice, and this process has been linked to increased cell survival signals in neurons [20]. Follow-up studies showed that transplantation of placenta-derived mitochondria via intravenous infusion significantly decreased brain infarction after focal cerebral ischemia in mice [79]. Similar neuroprotective effects were reported by Tashiro and colleagues [80], who found reduced neurological deficits following intravenous transplantation of astrocytic mitochondria in mice subjected to intracerebral haemorrhage. Pourmohammadi-Bejarpasi and colleagues also observed that intracerebroventricular transplantation of isolated mitochondria from human umbilical-cord-derived mesenchymal stem cells decreased apoptosis and gliosis, reduced infarct size, and improved motor function in a rat model of middle cerebral artery occlusion (MCAO) [81]. Babenko and colleagues reported that multipotent mesenchymal stem cells exhibited increased transport of functional mitochondria to astrocytes in an *in vitro* model of brain ischemia induced by oxygen-glucose deprivation [49]. This same study demonstrated that intravenous injection of multipotent mesenchymal stem cells (MMSC) in a rat model of experimental ischemic stroke reduced neurological deficits.

Similarly, intercellular mitochondria transfer has been identified as a key component mediating the therapeutic efficacy of previously identified treatments for stroke and traumatic brain injury (TBI), such as hyperbaric oxygen therapy (HBOT). For instance, transfer of mitochondria from astrocytes to injured neurons following HBOT was linked to reduced neuronal cell death in *in vitro* models of inflammation-plagued secondary cell death associated with stroke and TBI [82]. 

TBI is another condition for which intercellular mitochondrial transfer has been identified as a promising therapeutic strategy. Zhang and colleagues [83] reported that brain-tissue-derived mitochondria locally injected into the cortex reduced cellular apoptosis, attenuated blood–brain barrier leakage, and improved neurologic outcomes in a mouse model of TBI using a controlled cortical impact (CCI) device. Follow-up studies using mitochondria derived from liver directly injected into the cerebral cortex showed that mitochondria were internalised by astrocytes, microglia, and neurons and reduced neuronal apoptosis, attenuated anxiety, and improved spatial memory in a CCI model of TBI [84]. Additionally, mitochondria transplantation increased brain-derived neurotrophic factor (BDNF) expression in reactive astrocytes.

### 5.2. Neurodegenerative Diseases

Impairments in mitochondrial function are consistently observed in neurodegenerative diseases, including Alzheimer’s disease (AD), Parkinson’s disease (PD), and multiple sclerosis (MS). Accordingly, intercellular mitochondria transfer has been investigated as a potential therapeutic strategy in a wide range of neurodegenerative conditions. For instance, Nitzan et al. showed that intravenous injection of mitochondria isolated from humans reduced neuronal loss, decreased gliosis, and ameliorated cognitive deficits in a mouse model of AD (intracerebroventricularly injected with amyloid-β) [85]. 

Recently, Chang and colleagues [86] demonstrated the therapeutic potential of artificial mitochondrial transplantation *in vivo* in a PD rat model. In this study, mitochondria conjugated with the cell-penetrating peptide Pep-1 were injected into the medial forebrain bundle, where they were internalised by neurons and increased mitochondrial function and decreased dopaminergic neuron loss. Follow-up studies by the same group using intranasal delivery of mitochondria in PD rats reported that mitochondrial transplantation improved mitochondrial function, decreased neuronal loss, and attenuated PD-induced behavioural deficits [87]. Similarly, intravenously administered mitochondria isolated from human hepatoma cells decreased apoptosis and necrosis, reduced reactive oxygen species (ROS) levels, and attenuated behavioural impairments in a PD mouse model [88]. These findings were later replicated *in vitro*, in which induced-pluripotent stem cells (iPSCs)-derived astrocytes or astrocytic conditioned media reversed dopaminergic neuron degeneration and axonal pruning in a rotenone-induced PD model via mitochondrial transfer [89].

Similarly, Peruzzotti-Jametti and colleagues [56] investigated the effect of intracerebroventricular injection of neural stem cells (NSCs) or EVs into mice with myelin oligodendrocyte glycoprotein (MOG)_35-55_-induced chronic experimental autoimmune encephalomyelitis (EAE), an animal model of multiple sclerosis. NSCs actively transferred mitochondria to mononuclear phagocytes and astrocytes and ameliorated EAE disease severity, suggesting that mitochondrial transfer is a promising therapeutic strategy for the treatment of multiple sclerosis. 

### 5.3. Neurodevelopmental Disease

Mitochondrial dysfunction has been observed in neurodevelopmental conditions, such as schizophrenia (SZ) and fragile X syndrome (FXS), including altered mitochondrial dynamics, collapse of mitochondrial membrane potential, decreased ATP levels, increased ROS, and functional and transcriptional alterations in genes related to energy metabolism and oxidative stress [76,90,91,92,93,94,95].

Recently, Robicsek and colleagues [96] reported that transfer of isolated active normal mitochondria (IAN-MIT) attenuated impaired mitochondrial function in SZ-derived cells *in vitro*, including SZ-derived lymphoblasts, iPSCs, and H9c2 myoblasts. In SZ-lymphoblasts, IAN-MIT transfer normalized impaired basal respiration and recovered mitochondrial membrane potential in SZ-neurons. In addition, this same study investigated the effects of *in vivo* IAN-MIT transfer using the poly(I:C) rat model of SZ. Intra-prefrontal cortex injection of IAN-MIT to adolescent rats prenatally exposed to poly-I:C, attenuated decreased mitochondrial membrane potential, and prevented the emergence of SZ-like selective attention deficit in adulthood.

Fragile X syndrome (FXS) is an inherited disorder that is characterized by intellectual disability and is caused by the deficiency or absence of the *Fmr1* gene, which encodes fragile X mental retardation protein (FMRP), an RNA binding protein in neurons that is essential for synaptic plasticity. In a recent study, Ha and colleagues [97] reported that mitochondrial components, including nuclear respiratory factor 1 (NRF-1), ATP synthase F1 subunit α (ATP5A), ATP synthase F1 subunit β (ATPB), and voltage dependent anion channel 1 (VDAC1), were reduced in EVs secreted by astrocytes of *Fmr1* KO mice, a model of FXS. These reductions were accompanied by a decrease in mitochondrial biogenesis and mitochondrial membrane potential in astrocytes from Fmr1 KO mice, suggesting that EV-mediated transport of mitochondrial components may underlie FXS. However, whether EVs containing whole mitochondria for intercellular transfer are reduced in FXS is not known. Further the therapeutic efficacy of mitochondrial transfer was not tested in this model.

### 5.4. Chemotherapy

Chemotherapy-induced neurotoxicity is a serious health problem that occurs during or after chemotherapy. Several chemotherapy drugs have been shown to cross the blood–brain barrier at concentrations that are sufficient to cause neuronal damage [98]. Glial–neuronal mitochondrial transfer appears to exert neuroprotective effects in response to chemotherapy-induced neurotoxicity. For instance, mitochondrial transfer from astrocytes increased neuronal survival, restored neuronal mitochondrial membrane potential, and normalized neuronal calcium dynamics in neurons treated with the chemotherapeutic drug cisplatin [18]. Other studies have shown that mesenchymal stem cell (MSC)-derived mitochondrial transfer to neural stem cells reverses cisplatin-induced cell death and decreases mitochondrial membrane potential [50]. In addition, irradiation has been shown to have differing effects on TNT induction and mitochondria transfer in glioblastoma stem-like cells *in vitro* depending on the time course of treatment [36]. These findings have important implications for whether mitochondrial transfer should be used as therapeutic treatment in cancer, as it may be used by tumour cells as a rescue mechanism and evasion of apoptosis and tumour progression.

Taken together, these findings suggest that alterations in intercellular mitochondrial transfer are widely observed in brain injury and disease and that treatments aimed at enhancing mitochondrial transfer are promising therapeutic strategies.

## 6. Therapeutic Strategies Targeting Mitochondrial Transfer

Although intercellular mitochondrial transport has been shown to occur sporadically in the healthy brain as well as under disease conditions and following inflammatory stimulation, a number of strategies aimed at enhancing mitochondrial transfer have been proposed. In the following section, we discuss current research on pharmacologic approaches to enhancing intercellular mitochondrial transfer as well as mitochondrial transplantation strategies in the brain. Of note, this research may also inform strategies to impair the transfer of healthy mitochondria to cancerous cells during chemotherapy.

### 6.1. Pharmacologic Approaches

#### 6.1.1. CD38

CD38 (cluster of differentiation 38) is an enzyme that catalyses cyclic ADP-ribose (cADPR) activity and is primarily expressed in neurons, astrocytes, and microglia [99]. Recently, CD38 has been identified as a promising target for the enhancement of intercellular mitochondrial transfer both *in vitro* and *in vivo*. For instance, studies have shown that induction of CD38 signalling by cADPR stimulation increased the release of extracellular mitochondria from astrocytes *in vitro* (Figure 3) as well as the functionality of extracellular mitochondria [20]. Conversely, suppression of CD38 in astrocytes by short interfering RNA (siRNA) reduced mitochondrial transfer to adjacent neurons *in vitro* and decreased measures of neuronal health, such as dendrite regrowth [20]. In a disease-specific context, downregulation of CD38 by siRNA injection decreased astrocyte-to-neuron mitochondrial transfer and worsened neurological outcomes in a mouse model of focal cerebral ischaemia [20]. Similarly, pharmacologic inhibition of CD38 with quercetin or apigenin or genetic inhibition with CD38-targeting shRNA reduced mitochondrial transfer efficiency in human astrocyte and neuronal cell lines, whereas stimulation with cADPR promoted mitochondrial transfer [31]. Furthermore, mutations in the astrocytic protein GFAP have been shown to impair astrocytic mitochondrial transfer and lower CD38 mRNA expression levels [31]. However, overexpression of CD38 in these cells was unable to rescue the defect in mitochondrial transfer, suggesting that other mediators in addition to CD38 may be compromised in GFAP-mutated astrocytes.

CD38 has been implicated in both TNT formation [100] as well as EV formation [20], suggesting that CD38 may modulate different mechanisms of intercellular mitochondrial transfer. Neurons have been shown to play a key role in CD38 signalling in astrocytes, as neuronal release of glutamate induces CD68 overexpression in co-cultured astrocytes [101]. Mechanistically, CD38-cADPR signalling has been shown to boost extracellular mitochondrial functionality via post-translational O-GlcNAcylation [102]. O-GlcNAcylation is a post-translational modification that results in the addition of *O*-*GlcNAc* to Ser/Thr residues in nuclear and cytoplasmic proteins [103] and is thought to be involved in the regulation of mitochondrial function [104,105]. Activation of CD38-cADPR signalling in astrocytes was reported to induce protein O-GlcNAcylation in extracellular mitochondria, and O-GlcNAcylated mitochondria were more efficiently transferred to neurons where they improved neuroprotection against oxygen-glucose deprivation [102]. Taken together, these findings suggest that CD38 is a promising target for pharmacologic modulation of intercellular transport in the brain; however, further research is needed to elucidate its mechanism of action.

#### 6.1.2. MIRO-1

Mitochondrial Rho-GTPase 1 (Miro1) is an outer mitochondrial membrane protein that plays a crucial role in microtubule-based mitochondrial motility and has recently been identified as a key player in mitochondrial transfer via TNT formation (Figure 3) [106]. Miro-1 overexpression enhances the transfer of mitochondria from mesenchymal stem cells (MSCs) to neuronal stem cells *in vitro* [50]. Similarly, overexpression of Miro-1 in MSCs increased TNT formation and transfer of mitochondria to co-cultured astrocytes *in vitro* [49]. Conversely, siRNA-mediated knockdown of Miro-1 in astrocytes reduced mitochondrial transfer from astrocytes to neurons and prevented the normalization of neuronal calcium dynamics [18]. Similarly, mitochondrial transfer efficiency was reduced in astrocytes treated with shRNAs targeting Miro1 and Miro2, whereas overexpression of Miro1 increased mitochondrial transfer [31]. Thus, Miro1 is a promising target for enhancement of mitochondrial transfer although the mechanisms through which Miro1 acts remain to be elucidated.

#### 6.1.3. CX43

Astrocyte connexin 43 (Cx43) is a ga- junction-associated protein that has been identified as a potential therapeutic target for the enhancement of glial–neuronal mitochondrial transfer via TNTs. Cx43, which is encoded by the *GJA1* gene, has been shown to be involved in mitochondrial stabilization by up-regulating functional GJA1, which is necessary for mitochondrial movement [19]. Importantly, adeno-associated virus (AAV)-mediated overexpression of GJA1-20K, the most abundant isoform produced by GJA1 alternative translation, enhanced mitochondrial transfer from astrocytes to neurons *in vitro* [19]. Conversely, this same study showed that Cx43 knockdown with Gap26, a specific Cx43 hemichannel blocker, inhibited astrocyte-neuron mitochondrial transfer.

#### 6.1.4. Mitochondrial Fission/Fusion

Although the intercellular transfer of damaged mitochondria from neurons to astrocytes has been shown to aid in neuroprotection [31,32], transfer of damaged mitochondria from glia to neurons or between glia is thought to contribute to neurodegeneration. For instance, LPS-induced mitochondrial fission is known to contribute to the release of dysfunctional mitochondria from microglia, which in turn triggers inflammatory astrocytic phenotypes and propagates neurodegeneration [35]. Thus, therapeutic strategies aimed at enhancing the viability of extracellularly released mitochondria may attenuate potential detrimental effects of intercellular mitochondrial transfer in specific disease models. In line with this, treatment with P110, a selective peptide that inhibits dynamin-related protein 1/fusion 1 (DRP1/FIS1)-mediated mitochondrial fission, or with Mdivi-1, an inhibitor of the catalytic activity of Drp1, has been shown to enhance the integrity and function of extracellular mitochondria released from microglia [35] (Figure 3). P110 treatment in microglia also led to a reduction in neuronal cell death, triggered by exposure to damaged extracellular mitochondria from glial cells. 

On the other hand, damaged mitochondrial transfer may play an important role in signalling to adjacent cells to initiate an adaptive reparative response. For instance, the transfer of damaged mitochondria from impaired somatic cells and their subsequent uptake in MSCs has been shown to induce production of the cytoprotective enzyme heme oxygenase-1 (HO-1) and stimulation of mitochondrial biogenesis [107]. Subsequently, the capacity of MSCs to donate their healthy mitochondria to impaired somatic cells to combat oxidative stress injury was enhanced. Thus, further studies are warranted to investigate the therapeutic efficacy of pharmacologically targeting extracellular mitochondrial viability in the brain, in particular, whether it may interfere with the promotion of adaptive reparative responses in adjacent cells.

### 6.2. Mitochondrial Transplantation

Artificial mitochondrial transplantation (AMT) is a technique that was pioneered by Clark and Shay [108] in 1982 and involves the transfer of exogenous mitochondria from healthy cells to stressed cells to aid in cell recovery. AMT has been shown to confer protective effects in the liver, heart, and bone marrow *in vivo* [24,106,109,110,111], and these studies have sparked widespread interest in AMT as a therapeutic strategy for brain disorders. Accordingly, numerous studies have reported promising findings regarding the therapeutic effects of mitochondrial transfer in brain disease and injury. Specifically, mitochondrial transplantation has been shown to exert neuroprotective effects and ameliorate disease severity in a range of conditions, including neurodegenerative disorders such as AD [85], PD [86], and MS [56] as well as stroke [20], TBI [83], and neurodevelopmental disorders [96] (Table 2).

Nevertheless, the method of delivery of mitochondrial transplantation to the brain remains a challenge due to the difficulty of crossing the blood–brain barrier. Intracerebral injection (IC) is the most direct method for drug delivery to the brain (Figure 3). Accordingly, a number of studies using direct injection of exogenous mitochondria into the cortex [20,83,84,96], medial forebrain bundle [86], or local intracerebral injection [56,78,81] have reported that transplanted mitochondria were present in brain resident cells, where they exerted protective effects. However, these methods are invasive, and the translational value of these findings to human studies is limited due to the fact that tissue damage may occur. 

Less invasive methods, such as intravenous infusion [49,78,79,80,85,88] and intranasal delivery [87], have also reported that transplanted mitochondria are present in the brain and exert protective effects. However, specific or targeted delivery of transplanted mitochondria may be difficult to achieve in the brain via these methods due to uneven cellular distribution and uptake of mitochondria as well as the difficulty in targeting specific cell populations and disease-affected brain regions. For instance, several studies reported that intravenous infusion of mitochondria resulted in the presence of transplanted mitochondria in the liver [85], muscle [78], lung, kidney, and heart [79] in addition to the brain.

Few studies to date have compared the efficacy of different methods of mitochondrial transplantation in the brain. Of the few studies conducted, Huang and colleagues [78] compared the efficacy of IC versus intravenous infusion of xenogenic mitochondria in brain-ischemic rats. They reported that IC injection resulted in a significant recovery in motor activity at 7 days postoperation, whereas the rats that received intravenous infusion only showed equivalent levels of motor recovery 14 days postoperation (but not at 7 days). The cellular distribution of mitochondria also differed, with higher mitochondrial uptake observed in neurons and astrocytes following IC injection, whereas equivalent mitochondrial uptake was observed in microglia in both IC- and intravenous-treated rats. These findings suggest that the method of mitochondria transplantation impacts both the therapeutic efficacy and cellular distribution of mitochondria in mouse models of disease; however, additional studies comparing the efficacy of IC versus intravenous and intranasal delivery are needed to further characterise these differences.

The source of transplanted mitochondria may also impact therapeutic efficacy and uptake in brain injury and disease. Specifically, whether allogeneic (same species) or xenogeneic (different species) mitochondria are used may produce differing results. For instance, Chang and colleagues [86] reported that while both xenogeneic and allogeneic mitochondria conferred protective effects in a PD mouse model, xenogeneic mitochondria transplantation was less effective at 3 months postoperation compared to allogenic transplantation. Similarly, mitochondria derived from different tissue or cell sources may have different donor properties. In line with this, Paliwal and colleagues [112] showed that human MSCs derived from bone marrow, adipose, dental pulp, and Wharton’s jelly differed in their donation capacity as well as their ability to suppress ROS levels *in vitro*.

Although AMT has been shown to exert protective effects on acute measures of brain injury and disease, its effects on long term measures are unclear. For instance, whereas mitochondrial transplantation contributed to the maintenance of normal bioenergetics acutely in injured spinal cord tissue, no effect was observed on long-term functional outcome measures, such as tissue sparing or restoration of motor and sensory functions [113]. Furthermore, most studies to date investigating the effects of mitochondrial transplantation in the brain have only confirmed the presence of transplanted mitochondria in brain resident cells for up to 12 weeks [86]. Future studies investigating the longitudinal effects of AMT in the brain are needed to confirm the efficacy and safety of this technique. In addition, the use of stem cells for AMT may increase the risk of developing tumours, as secondary cancers have been identified as a late complication of stem cell transplantation in humans [114].

In addition, a number of agents that enhance mitochondrial transfer and uptake have been identified that may boost the therapeutic efficacy of AMT. For instance, the cell-penetrating peptide Pep-1 was recently described as a powerful agent for the delivery of mitochondria to cells due to its ability to translocate various proteins and molecules across biological membranes [115]. Chang and colleagues [86] demonstrated that Pep-1-conjugated mitochondria injected into the medial forebrain bundle increased mitochondrial function and decreased dopaminergic neuron loss in a PD rat model. Pep-1-mediated mitochondrial delivery was also shown to restore mitochondrial function *in vitro* in a model of myoclonic epilepsy [116].

Finally, few studies to date have investigated the efficacy of AMT in human patients. Although Emani and colleagues [117] reported protective effects following transplantation of mitochondria into the myocardium of human paediatric patients with myocardial ischemia, with no adverse short-term complications [117], there have been no studies to date investigating AMT in the brain of human patients. As current research investigating the therapeutic efficacy of AMT in the brain is limited to pre-clinical animal studies, further research is needed to verify the safety and efficacy of this method for human patients.

**Table 2 cells-11-03603-t002:** In Vivo Artificial Mitochondrial Transplantation in Brain Injury and Disease.

Disease Model	Source of Mitochondria	Method of Delivery for AMT	Effects	References
Stroke	MMSC with overexpressed Miro-1	I.V. injection	-Increased neurological function	[49]
-MCAO model of focal ischemia				
Stroke	Primary mouse astrocytes	Local injection into peri-infarct cortex	-Upregulation of cell survival signals	[20]
-Focal cerebral ischaemia				
Stroke	Baby hamster kidney fibroblast (BHK-21)	ICV or systemic intra-arterial injection	-Increased motor performance	[78]
-MCAO model of focal ischemia			-Decreased brain infarct area	
			-Decreased neuronal death	
Stroke	Mouse placenta	I.V. injection	-Decreased brain infarct area	[79]
-Focal cerebral ischaemia				
Stroke	Primary mouse astrocytes	I.V. injection	-Increased neuronal viability	[80]
-Intracerebral haemorrhage			-Reduced neurologic deficits	
			-Restored Mn-SOD levels	
Stroke	Human umbilical-cord-derived mesenchymal stem cells	ICV	-Decreased apoptosis	[81]
-MCAO model of focal ischemia			-Decreased gliosis	
			-Improved motor function	
			-Decreased brain infarct area	
Chemotherapy-induced neurotoxicity	MSC	Intranasal	-Reduced apoptosis	[50]
-Cisplatin treatment				
Alzheimer’s disease	HeLa cells	I.V. injection	-Improved cognitive function	[85]
-Amyloid-β intracerebroventricularly injected			-Decreased neuronal loss	
			-Decreased gliosis	
			-Increased citrate-synthase and cytochrome c oxidase activities	
Parkinson’s disease	PC12 cells	Local injection into MFB	-Improved locomotive activity	[86]
-6-OHDA-lesioned rat model	or		-Increased neuronal survival	
	Human osteosarcoma cybrids		-Restored mitochondrial dynamics	
Parkinson’s disease	Rat liver	Intranasal	-Improved locomotive activity	[87]
-6-OHDA-lesioned rat model			-Increased neuronal survival	
			-Decreased oxidative damage	
Parkinson’s disease	HepG2 cells	I.V. injection	-Improved locomotive activity	[88]
-MPTP-induced mouse model			-Increased ATP levels	
			-Decreased ROS levels	
Multiple sclerosis	Neural stem cells	ICV	-Ameliorated EAE severity	[56]
-MOG_35-55_-induced EAE				
Schizophrenia	Human lymphoblasts	Intra-prefrontal cortex injection	-Rescued attentional deficits	[96]
-Prenatal poly-I:C exposure	Or		-Increased MMP	
	Rat brain			
Traumatic brain injury	Mouse liver	Local injection into cerebral cortex	-Increased ATP levels	[84]
-Controlled cortical impact	Mouse muscle		-Upregulated astrocytic BDNF	
			Improved spatial memory and cognitive function	
Traumatic brain injury	Mouse brain	Local injection into cerebral cortex	-Decreased apoptosis	[83]
Controlled cortical impact			-Increased angiogenesis	
			-Decreased brain oedema	
			-Decreased blood brain barrier leakage	

Abbreviations: MCAO, middle cerebral artery occlusion; 6-OHDA, 6-hydroxydopamine; MPTP, 1-methyl-4-phenyl-1,2,3,6-tetrahydropyridine; MMSC, multipotent mesenchymal stem cell; MSC, mesenchymal stem cell; I.V., intravenous; MFB, medial forebrain bundle; ICV, intracerebroventricularly; MOG_33-55_, myelin oligodendrocyte glycoprotein; EAE, experimental autoimmune encephalomyelitis; BDNF, brain-derived neurotrophic factor; AMT, artificial mitochondrial transplantation.

## 7. Conclusions and Future Directions

Intercellular mitochondrial transfer is a promising field of study for the advancement of mitochondria-targeted therapeutics in brain injury and disease. In particular, intercellular mitochondrial transfer can overcome the limitations of previously identified mitochondria-targeted therapeutics by offering a wholesale approach to addressing multifaceted aspects of mitochondrial dysfunction. As described in this review, mitochondria transfer between glia and neurons in the brain plays a crucial role in enhancing cell viability, aiding in the degradation of dysfunctional mitochondria, and modulating glia-mediated neuroinflammation. In addition, AMT has been shown to confer neuroprotective effects in pre-clinical animal models of brain injury, neurodegenerative disease, and neurodevelopmental disorders. These findings suggest that intercellular mitochondria transfer is an attractive target for the treatment a range of neurological disorders; however, further research is needed to elucidate the safety, efficacy, and mechanistic pathways underlying this process in human patients.

## Figures and Tables

**Figure 1 cells-11-03603-f001:**
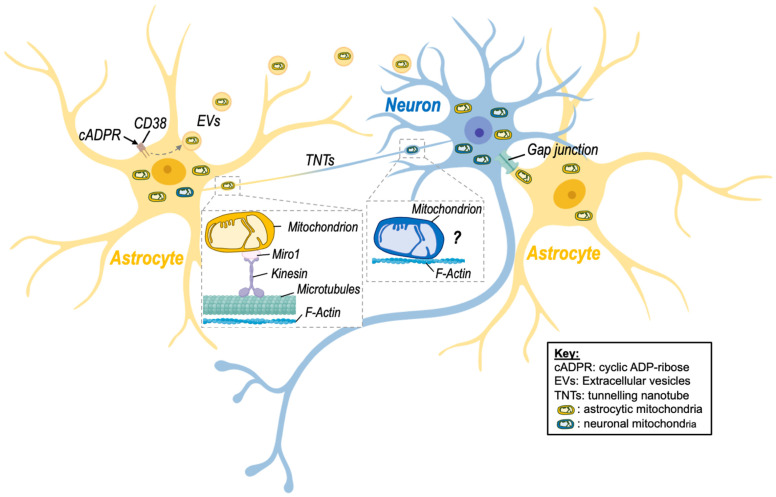
Intercellular mitochondrial transfer between neurons and astrocytes. Evidence of mitochondrial transfer between neurons and astrocytes has been described, and different processes have been highlighted. Namely, activation of CD38 (cluster of differentiation 38) by cyclic ADP-ribose (cADPR) induced the release of extracellular vesicles (EVs)-containing mitochondria by astrocytes in a model of stroke, improving neuronal viability. Transfer of mitochondria via tunnelling nanotubes (TNTs) has also been shown. This mode of transfer seems to rely on components of the cytoskeleton (microtubules and F-actin) as well as on the mitochondrial Rho-GTPase-1 protein (Miro-1) and its connection with kinesin for mitochondrial transport. Although transfer of neuronal mitochondria via TNTs is depicted on the figure, there is no clear evidence that mitochondria can be transferred from neurons to astrocytes via this route. Transfer of mitochondria between neuronal cells might involve microtubule-independent mechanism that remain to be identified. Nevertheless, transfer of astrocytic mitochondria to neurons via TNTs has been described. Other processes involving gap junctions may also be involved but still need to be studied in more detail in the brain. Created with BioRender.com (accessed on 7 November 2022).

**Figure 2 cells-11-03603-f002:**
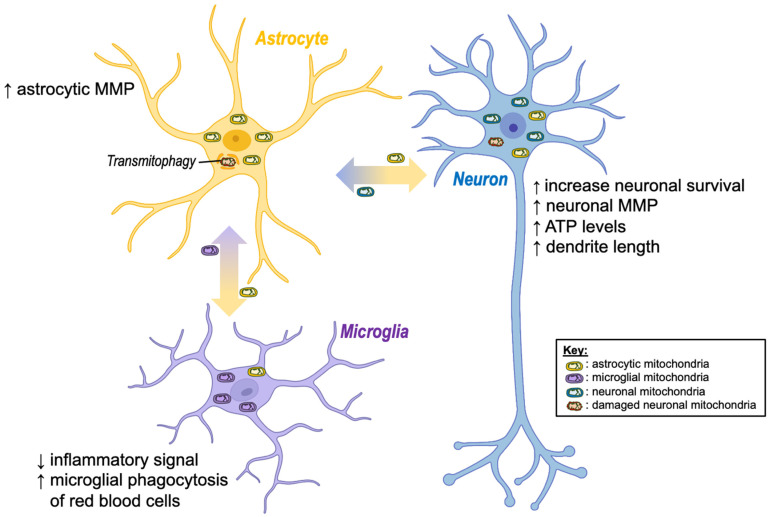
Beneficial effects of intercellular mitochondria transfer in the brain. Regardless of the mode of transfer (extracellular vesicles or tunnelling nanotubes), exchange of mitochondria between brain cells has shown positive effects. Namely, the transfer of healthy astrocytic mitochondria to stressed neurons (e.g., after a stroke) seems to increase neuronal viability, dendrite outgrowth, and bioenergetics (ATP, MMP), while damaged neuronal mitochondria are degraded by astrocytes. Moreover, transfer of mitochondria from astrocytes to microglia appears to reduce inflammation and increase microglial phagocytosis. ↓ decrease; *↑* increase; ATP, adenosine triphosphate; MMP, mitochondrial membrane potential. Created with BioRender.com (accessed on 7 November 2022).

**Figure 3 cells-11-03603-f003:**
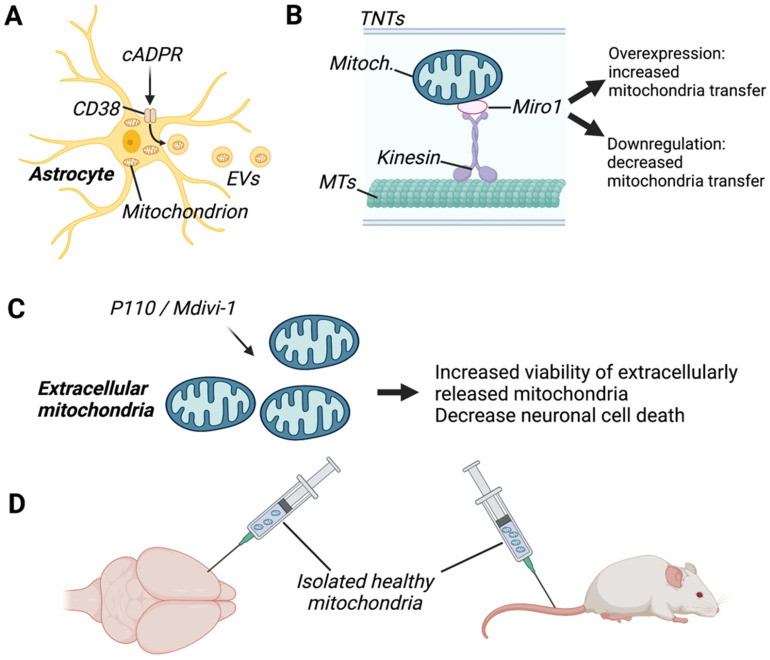
Examples of therapeutic approaches targeting mitochondrial transfer in the brain. (**A**) Enhancement of mitochondrial transfer via extracellular vesicles (EVs) was performed by activating astrocytic CD38 (cluster of differentiation 38) with cyclic ADP-ribose (cADPR), which improved neuronal viability in a model of stroke. (**B**) Enhancement of mitochondrial transfer via tunnelling nanotubes (TNTs) was done by overexpressing the mitochondrial Rho-GTPase 1 (Miro1), which increased TNT formation. (**C**) Inhibition of mitochondrial fusion proteins and dynamin-related protein 1/fusion 1 (DRP1/FIS1), with P110 or Mdivi1, was shown to improve the viability of extracellular mitochondria, decreasing neuronal death. (**D**) Therapeutic approaches consisting of intracranial or intravenous injection of healthy mitochondria in rodents revealed that transplanted mitochondria are present in brain resident cells and exerted protective effects. MTs, microtubules. Created with BioRender.com (accessed on 7 November 2022).

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
