# Peer review of "Mitochondria Transfer in Brain Injury and Disease"

_cells, 2022, doi:10.3390/cells11223603_

Round 1

Reviewer 1 Report

In the review by Fairley et al, titled "Glia-Neuronal Mitochondrial Transfer in Injury and Disease" the authors summarize the key points relevant to the transfer of healthy or altered mitochondria between glia and neuronal cells as a neuroprotective mechanism.

Below my specific comments.

Paragraph 2: "Evidence for intercellular transfer of mitochondria in the brain":

Major comments:

In this section the authors list the different exchanges of mitochondria between different brain cell types described in the literature. However, they do not specify under what conditions this transfer occurs such as: stress conditions, disease, treatment, etc. Neither the state of the mitochondria, and more important the benefits that the interchange of mitochondria confers to the donor or recipient population. Finally, an important missing part in this paragraph is the transfer of mitochondria in brain cancer.

Minor comments:

The end of the paragraph about the reduced ability of oligodendrocytes to internalize mitochondria does not connect with the rest of the paragraph.

Detailed comments:

-        State and fate of mitochondria: Only in line 86 they mention the transfer of disrupted mitochondria from neurons to astrocytes for disposal. They should also add the reference of Davis et al, 2014 10.1073/pnas.1404651111. And also more recent paper from Lampinen et al, 2022 "Neuron-astrocyte transmitophagy is altered in Alzheimer's disease"10.1016/j.nbd.2022.105753

Furthermore, the transfer of healthy mitochondria from astrocytes to neurons has been described as a neuroprotective mechanism in Hayakawa, et al 2016 10.1038/nature18928.

-        Transfer of mitochondria in Glioblastoma

It has been described the transfer of mitochondria in brain tumor between tumor-to-tumor cell and between tumor cell-to-tumor microenvironment cell.

Between tumor-to-tumor cells: In glioblastoma stem-like cells these transfer of mitochondria has been correlated with different responds of the glioblastoma stem-like cells to irradiation treatment Pinto, et al 2020 10.1042/BCJ20200710.

Between tumor and tumor microenvironment:

Interchange between glioblastoma and non-tumoral astrocytes: which can have a astrocytic protection role under hypoxic conditions Valdebenito et al 2021 10.1038/s41598-021-93775-8. And the association on delivery non-neoplastic mitochondria on glioblastoma drug response as well as proliferation and migration Civita et al, 2019 10.3390/ijms20236017.

Paragraph 3: "Structural Mechanisms of Mitochondrial Transfer"

I suggest adding a table summarizing the disease in which mitochondria transfer via TNTs or EVs has been described, the status of the mitochondria, the consequence of the transfer if known, and the respective references.

-        Extracellular Vesicles:

Again, the authors must specify under what conditions the EVs with mitochondria inside are delivered (stress conditions, diseases, etc.) and the consequences of the uptake of those EVs in the acceptor population.

Line 106.   For instance, the small EVs Secreted by Nigrostriatal Astrocytes Rescue Cell Death and Preserve Mitochondrial Function in Parkinson's Disease (Leggio et al, 2022 doi.org/10.1002/adhm.202201203).

-        Tunneling nanotubes:

Major comments

This paragraph should be improved, the description of the Tunneling nanotubes is not accurate and the references are not the more relevant in this field. Furthermore, they should  mention the transfer of mitochondria via TNTs in different disease and the consequences, for example: the transfer of mitochondria via TNTs have been described to rescues apoptotic PC12 (Wang et Gerdes, 2015 https://doi.org/10.1038/cdd.2014.211)

Detailed comments:

-        They should mention the first time that this type of communication was described in by the group of Gerdes (Rustom et al, 2004. 10.1126/science.1093133).

-        The definition "Tunneling nanotubes (TNTs) are membranous channels comprised of F-actin that connect two or more cells and transfer cytoplasmic molecules from one cell to another" is not complete because TNTs don’t only transfer cytoplasmatic molecules, they are able to transfer viruses, bacteria, and prions which use them as a route of propagation of the disease (Pepe A et al, 2022 10.1126/sciadv.abo0171, Onfelt et all 2006 10.4049/jimmunol.177.12.8476; Gousset et al. 2009 10.1038/ncb1841). Furthermore, they can transfer nanoparticles that could be use in the future as a nanocarrier mechanism of drug delivery (Saenz-de-Santa-Maria I et al, 2017 10.18632/oncotarget.15467). I would therefore specifically mention transfer of organelles such as mitochondria instead of "transfer cytoplasmic molecules" this is very ambiguous. This will be more coherent with the title of the section.

-        Other Mechanisms: As the authors say the transfer of mitochondria via gap junctions or cell fusions needs further investigations and it is not clear.

Figure 1: Intercellular mitochondrial transfer between neurons and astrocytes. Scheme of the different mechanism proposed for mitochondrial transfer between neurons and astrocytes. The cartoon of the figure is confusing. It looks like the mitochondria transfer occurs only via EVs or gap junctions. The authors should draw a TNT connecting the astrocyte and the neuron.

As an example of TNTs mediated transfer of mitochondria, the authors made a magnification of a single yellow mitochondrion (astrocytic mitochondria) that is hovering around the cells and draw the microtubules motor proteins that could be implicated in the transfer of mitochondrion via TNTs. This is incomplete because the motor proteins implicated in the transfer of mitochondria via TNTs are yet unknown.

In my opinion they should add the possibility of transfer neuronal mitochondrion (blue mitochondrion) from the neuron to the astrocytes. In that case, the proposed motor proteins (miro1/2, Milton and Kinesin), cannot be implicated in the transfer of mitochondria through TNTs formed by the neurons because they only contain actin filaments, as shown in the case  of neuronal-like cells such as CAD cells and human neuroblastoma SH-SY5Y cells have TNTs only actin positive without microtubules (Sartori-Rupp et al, 2019 10.1038/s41467-018-08178-7). Therefore, they should add other candidates such as actin driven motor proteins: This should be  also described in the text, with the corresponding references.

Paragraph 4:  "Effects of Mitochondrial Transfer in the Brain "

4.1. Enhancement of Cell Viability

The authors named the correct references, but they should discuss more in depth the work of Hayakawa et al, 2016 Transfer of mitochondria from astrocytes to neurons after stroke https://doi.org/10.1038/nature18928 and the work of English et al, 2020 as astrocytes transfer healthy mitochondria to neurons after cisplatin treatment (https://doi.org/10.1186/s40478-020-00897-7 ).

4.2. Enhancement of Mitochondrial Degradation

Another example that they don’t mention is the important work of Sharma et al, 2019 that described how Rhes protein is able to travel via TNTs between striatal neuronal cell line from a healthy cell to the neighboring cells.

Figure2: Beneficial effects of intercellular mitochondria transfer in the brain.

The idea of the figure is good, but they should add the name of the cells next to the drawing- astrocytes next to the yellow cell, microglia the violet cells, and neuron blue cell.

4.4. Deleterious Effects

They should add some reference about the mitochondria hijacked in neuronal cell under infection with SARS-CoV-2.

Paragraph 5: "Mitochondrial Transfer in Brain Injury and Disease "

5.4 Chemotherapy

They should add also in this section the effect of radiotherapy, and cite/discuss the work of  Pinto et al, 2020 https://doi.org/10.1042/BCJ20200710

Paragraph 6: " Therapeutic Strategies Targeting Mitochondrial Transfer "

A missing possible therapeutic strategy directed to induce the mitophagy of disrupted mitochondria in astrocytes as a mechanism of clearance.

Furthermore, I would add a paragraph about therapeutic strategics to impair the transfer of healthy mitochondria in cancer as a mechanism of rescue or evasion of the apoptosis.

Paragraph 7: "Conclusions and Future Directions "

The authors develop the idea of the neuroprotective aspect of mitochondria transfer from glial cells to neurons and mitochondrial transplantation as a treatment for neurodegenerative diseases and brain injury. However, they have not considered the degradation of dysfunctional mitochondria by astrocytes which is key as a neuroprotective mechanism, and the development of therapeutic strategies to increase mitophagy of dysfunctional mitochondria. Most importantly, it is not considered how in cancer mitochondrial transfer can be used by tumor cells as a rescue mechanism, evasion of apoptosis and tumor progression. I believe that these ideas should be developed  in this section of the review.

Author Response

In the review by Fairley et al, titled "Glia-Neuronal Mitochondrial Transfer in Injury and Disease" the authors summarize the key points relevant to the transfer of healthy or altered mitochondria between glia and neuronal cells as a neuroprotective mechanism.

Below my specific comments.

Paragraph 2: "Evidence for intercellular transfer of mitochondria in the brain":

Major comments:

In this section the authors list the different exchanges of mitochondria between different brain cell types described in the literature. However, they do not specify under what conditions this transfer occurs such as: stress conditions, disease, treatment, etc. Neither the state of the mitochondria, and more important the benefits that the interchange of mitochondria confers to the donor or recipient population. Finally, an important missing part in this paragraph is the transfer of mitochondria in brain cancer.

Answer: We thank the referee for this comment. These points are addressed in later sections of the manuscript.

Namely, the conditions under which mitochondria transfer occurs is discussed in detail in sections:

3.2: The effects of stress on TNT formation are discussed

4.3: The effects of inflammatory stimuli on mitochondria transfer are discussed

5: The effects of different disease conditions on enhancing or inhibiting mitochondria transfer are discussed

The benefits of mitochondria transfer in recipient cells, and brain diseases is discussed in detail in sections:

4.1: Discusses how mitochondria transfer enhances cell viability in recipient cells

4.2: Discusses how transmitophagy enhances cell viability in donor cells

5: Discusses how mitochondria has been used as a therapeutic in a range of brain diseases and the beneficial outcomes

Mitochondria transfer in brain cancer is now discussed in section 2, lines 93-96 as follows:

Mitochondria transfer has also been observed between tumor-to-tumor cells, and tumor-to-tumor microenvironment in the brain. For instance, mitochondria transfer has been observed between glioblastoma stem-like cells in vitro [36], as well as between glioblastoma cells and surrounding non-tumor astrocytes [37].

Minor comments:

The end of the paragraph about the reduced ability of oligodendrocytes to internalize mitochondria does not connect with the rest of the paragraph.

Answer: We agree with the referee. A linking sentence has been added to better connect this sentence with the rest of the paragraph as follows:

lines 96-100 “Fewer studies have investigated mitochondrial transfer in oligodendrocytes in the brain, however, Zhao and colleagues [34] reported that oligodendrocytes exhibit reduced capacity to internalize exogenous mitochondria, with less than 10% of oligodendrocytes exhibiting internalized mitochondria. Thus, they are not discussed in detail in this review.

Detailed comments:

-        State and fate of mitochondria: Only in line 86 they mention the transfer of disrupted mitochondria from neurons to astrocytes for disposal. They should also add the reference of Davis et al, 2014 10.1073/pnas.1404651111. And also more recent paper from Lampinen et al, 2022 "Neuron-astrocyte transmitophagy is altered in Alzheimer's disease"10.1016/j.nbd.2022.105753

Answer: We thank the referee for this comment. Both references have now been added. Please see lines 88.

Furthermore, the transfer of healthy mitochondria from astrocytes to neurons has been described as a neuroprotective mechanism in Hayakawa, et al 2016 10.1038/nature18928.

Answer: The paper by Hayakawa et al. 2016 is mentioned twice in this paragraph (lines 82 & 87).

This paper is also discussed in more detail in later sections, including lines 182-188, 314-317, 433-466.

-        Transfer of mitochondria in Glioblastoma

It has been described the transfer of mitochondria in brain tumor between tumor-to-tumor cell and between tumor cell-to-tumor microenvironment cell.

Between tumor-to-tumor cells: In glioblastoma stem-like cells these this transfer of mitochondria has been correlated with different responds of the glioblastoma stem-like cells to irradiation treatment Pinto, et al 2020 10.1042/BCJ20200710. 

Between tumor and tumor microenvironment:

Interchange between glioblastoma and non-tumoral astrocytes: which can have a astrocytic protection role under hypoxic conditions Valdebenito et al 2021 10.1038/s41598-021-93775-8. And the association on delivery non-neoplastic mitochondria on glioblastoma drug response as well as proliferation and migration Civita et al, 2019 10.3390/ijms20236017.

Answer: We thank the referee for making us aware of this point. Transfer of mitochondria in glioblastoma has now been discussed and the relevant references added in section 2, lines 93-96 as follows:

“Mitochondria transfer has also been observed between tumor-to-tumor cells, and tumor-to-tumor microenvironment in the brain. For instance, mitochondria transfer has been observed between glioblastoma stem-like cells in vitro [36], as well as between glioblastoma cells and surrounding non-tumor astrocytes [37].”

Paragraph 3: "Structural Mechanisms of Mitochondrial Transfer"

I suggest adding a table summarizing the disease in which mitochondria transfer via TNTs or EVs has been described, the status of the mitochondria, the consequence of the transfer if known, and the respective references.

Answer: We agree with the referee. Two tables have been added to the manuscript. Table 1 summarizes the mechanisms of transfer used by cells in the brain (EVs or TNTs). Table 2 summarizes the various brain diseases in which mitochondria transfer has been investigated, and its therapeutic effects.

-        Extracellular Vesicles:

Again, the authors must specify under what conditions the EVs with mitochondria inside are delivered (stress conditions, diseases, etc.) and the consequences of the uptake of those EVs in the acceptor population. 

Line 106.   For instance, the small EVs Secreted by Nigrostriatal Astrocytes Rescue Cell Death and Preserve Mitochondrial Function in Parkinson's Disease (Leggio et al, 2022 doi.org/10.1002/adhm.202201203).

Answer: We thank the referee for this comment. A table has been added (table 1, line 168) summarizing the conditions in which EV-mediated mitochondria transfer occurs in the brain, and the consequences of their uptake.

The paper by Leggio et al. suggested by the reviewer was not added or discussed in the manuscript, because the EVs observed in this study did not contain mitochondria.

-        Tunneling nanotubes:

Major comments

This paragraph should be improved, the description of the Tunneling nanotubes is not accurate and the references are not the more relevant in this field. Furthermore, they should  mention the transfer of mitochondria via TNTs in different disease and the consequences, for example: the transfer of mitochondria via TNTs have been described to rescues apoptotic PC12 (Wang et Gerdes, 2015 https://doi.org/10.1038/cdd.2014.211)

Answer: We thank the referee for this comment. A table has been added (table 1, line 168) summarizing the conditions in which TNT-mediated mitochondria transfer occurs in the brain, and the consequences of their uptake.

The paragraph has been edited to include the suggestions of the reviewer as outlined in the next two comments.

The paper by Wang and Gerdes, 2015 is referenced in the table.

Detailed comments:

-        They should mention the first time that this type of communication was described in by the group of Gerdes (Rustom et al, 2004. 10.1126/science.1093133). 

Answer: We agree. The sections have been updated lines 120-122 as follows: “Tunneling nanotubes (TNTs) were first described in 2004 by Rustom and colleagues [42], who identified membranous channels comprised of F-actin that connect two or more cells and are capable of transferring mitochondria from one cell to another [43].”

-        The definition "Tunneling nanotubes (TNTs) are membranous channels comprised of F-actin that connect two or more cells and transfer cytoplasmic molecules from one cell to another" is not complete because TNTs don’t only transfer cytoplasmatic molecules, they are able to transfer viruses, bacteria, and prions which use them as a route of propagation of the disease (Pepe A et al, 2022 10.1126/sciadv.abo0171, Onfelt et all 2006 10.4049/jimmunol.177.12.8476; Gousset et al. 2009 10.1038/ncb1841). Furthermore, they can transfer nanoparticles that could be use in the future as a nanocarrier mechanism of drug delivery (Saenz-de-Santa-Maria I et al, 2017 10.18632/oncotarget.15467). I would therefore specifically mention transfer of organelles such as mitochondria instead of "transfer cytoplasmic molecules" this is very ambiguous. This will be more coherent with the title of the section.

Answer: We agree with the Reviewer’s comments and have changed the section as follows:

“Tunneling nanotubes (TNTs) were first described in 2004 by Rustom and colleagues [42], who identified membranous channels comprised of F-actin that connect two or more cells and are capable of transferring mitochondria from one cell to another [43].”

-        Other Mechanisms: As the authors say the transfer of mitochondria via gap junctions or cell fusions needs further investigations and it is not clear.

Answer: We thank the referee for this comment. This paragraph simply outlines that although other mechanisms of mitochondria transfer (gap junctions, cell fusion) have been identified in cell types outside of the brain, these methods of transfer have not yet been shown in brain cells. Thus, further investigation is required to elucidate whether these transfer mechanisms also occur in brain cells. We feel this point is clearly made, and so it has not been edited.

Figure 1: Intercellular mitochondrial transfer between neurons and astrocytes. Scheme of the different mechanism proposed for mitochondrial transfer between neurons and astrocytes. The cartoon of the figure is confusing. It looks like the mitochondria transfer occurs only via EVs or gap junctions. The authors should draw a TNT connecting the astrocyte and the neuron.

As an example of TNTs mediated transfer of mitochondria, the authors made a magnification of a single yellow mitochondrion (astrocytic mitochondria) that is hovering around the cells and draw the microtubules motor proteins that could be implicated in the transfer of mitochondrion via TNTs. This is incomplete because the motor proteins implicated in the transfer of mitochondria via TNTs are yet unknown. 

In my opinion they should add the possibility of transfer neuronal mitochondrion (blue mitochondrion) from the neuron to the astrocytes. In that case, the proposed motor proteins (miro1/2, Milton and Kinesin), cannot be implicated in the transfer of mitochondria through TNTs formed by the neurons because they only contain actin filaments, as shown in the case of neuronal-like cells such as CAD cells and human neuroblastoma SH-SY5Y cells have TNTs only actin positive without microtubules (Sartori-Rupp et al, 2019 10.1038/s41467-018-08178-7). Therefore, they should add other candidates such as actin driven motor proteins: This should be also described in the text, with the corresponding references.

Answer: We thank the reviewer for making us aware of this point. We have realized that there were discrepancies between the figures in the word file and the pdf (probably due to issues with the file conversion for .doc to .pdf). Therefore, some items present in the figure 1 disappeared, namely the TNTs. The problem is fixed now, and the image was modified according to the reviewer comment. We apologize for the circumstances.

Regarding the motor proteins implicated in the transfer of mitochondria via TNTs, the Miro-1 protein was clearly identified in a study focused on mitochondrial transfer between neurons and astrocytes (ref. 97). Indeed, they showed that knocking down Miro-1 in astrocytes reduced mitochondrial transfer from astrocytes to neurons. This study was mentioned in the text lines 468-480.

We also mention the study from Sartori-Rupp and colleagues which states that TNTs between neuronal cells are only composed of F-actin. Please see lines 138-141:

“Of note, Sartori-Rupp and colleagues found that mitochondria can be transported be-tween neuronal cells via individual TNTs composed of a single continuous bundle of parallel actin filaments [48]. This suggest that mitochondria directly bind to actin filaments and are transported via a microtubule- independent mechanism that remain to identify.”

Paragraph 4:  "Effects of Mitochondrial Transfer in the Brain "

4.1. Enhancement of Cell Viability

The authors named the correct references, but they should discuss more in depth the work of Hayakawa et al, 2016 Transfer of mitochondria from astrocytes to neurons after stroke https://doi.org/10.1038/nature18928 and the work of English et al, 2020 as astrocytes transfer healthy mitochondria to neurons after cisplatin treatment (https://doi.org/10.1186/s40478-020-00897-7 ).

Answer: These papers are discussed in greater detail in the section “Mitochondria Transfer in Brain Injury and Disease” as follows:

Lines 314-317: “Hayakawa and colleagues showed that an influx of astrocytic mitochondria into adjacent neurons is observed following induction of focal cerebral ischaemia in mice, and this process has been linked to increased cell survival signals in neurons [20].”

And

Lines 409-412: “For instance, mitochondrial transfer from astrocytes increased neuronal survival, restored neuronal mitochondrial membrane potential, and normalized neuronal calcium dynamics in neurons treated with the chemotherapeutic drug cisplatin [18].”

4.2. Enhancement of Mitochondrial Degradation

Another example that they don’t mention is the important work of Sharma et al, 2019 that described how Rhes protein is able to travel via TNTs between striatal neuronal cell line from a healthy cell to the neighboring cells.

Answer: This paper has now been discussed lines 208-212 as follows:

“In addition, Rhes protein, a critical regulator of mitophagy in the brain, has been shown to transfer between striatal neuronal cells via TNTs, where it binds to damaged mitochondria in the recipient cell, suggesting that neurons may also transfer mitophagy-enhancing proteins to aid the transmitophagy process [57].”

Figure2: Beneficial effects of intercellular mitochondria transfer in the brain.

The idea of the figure is good, but they should add the name of the cells next to the drawing- astrocytes next to the yellow cell, microglia the violet cells, and neuron blue cell. 

Answer: We thank the reviewer for making us aware of this point. As stated for the Figure 1, we have realized that there were discrepancies between the figures in the word file and the pdf (probably due to issues with the file conversion for .doc to .pdf). The problem is fixed now.

4.4. Deleterious Effects

They should add some reference about the mitochondria hijacked in neuronal cell under infection with SARS-CoV-2.

Answer: This reference has been added in the following section, lines 289-292:

“These findings also have implications for other disease-related cargo that are known to bind to mitochondria, such as SARS-CoV-2 [67], and suggest that under specific disease conditions, mitochondria may be hijacked as a means to facilitate cell-to-cell transfer of pathogens in the brain.”

Paragraph 5: "Mitochondrial Transfer in Brain Injury and Disease "

5.4 Chemotherapy

They should add also in this section the effect of radiotherapy, and cite/discuss the work of  Pinto et al, 2020 https://doi.org/10.1042/BCJ20200710

Answer: This paper has now been discussed lines 414-417 as follows:

“In addition, irradiation has been shown to have differing effects on TNT induction and mitochondria transfer in glioblastoma stem-like cells in vitro, depending on the time-course of treatment [36].”

Paragraph 6: " Therapeutic Strategies Targeting Mitochondrial Transfer "

A missing possible therapeutic strategy directed to induce the mitophagy of disrupted mitochondria in astrocytes as a mechanism of clearance.

Furthermore, I would add a paragraph about therapeutic strategics to impair the transfer of healthy mitochondria in cancer as a mechanism of rescue or evasion of the apoptosis.

Answer: The therapeutic potential of impairing mitochondria transfer to cancerous cells is now mentioned as follows:

Lines 417-419: “These findings have important implications for whether mitochondrial transfer should be used as therapeutic treatment in cancer, as it may be used by tumor cells as a rescue mechanism, for evasion of apoptosis and tumor progression.”

Lines 429-430: “Of note, this research may also inform strategies to impair the transfer of healthy mitochondria to cancerous cells during chemotherapy.”

Paragraph 7: "Conclusions and Future Directions "

The authors develop the idea of the neuroprotective aspect of mitochondria transfer from glial cells to neurons and mitochondrial transplantation as a treatment for neurodegenerative diseases and brain injury. However, they have not considered the degradation of dysfunctional mitochondria by astrocytes which is key as a neuroprotective mechanism, and the development of therapeutic strategies to increase mitophagy of dysfunctional mitochondria. Most importantly, it is not considered how in cancer mitochondrial transfer can be used by tumor cells as a rescue mechanism, evasion of apoptosis and tumor progression. I believe that these ideas should be developed  in this section of the review.

Answer: The degradation of dysfunctional mitochondria is mentioned in line 620.

The effects of mitochondria transfer in cancer is now addressed as follows:

Lines 416-419: “These findings have important implications for whether mitochondrial transfer should be used as therapeutic treatment in cancer, as it may be used by tumor cells as a rescue mechanism, evasion of apoptosis and tumor progression.”

Reviewer 2 Report

IIn the present review article, the authors have well described the connection between neurons and glia during pathological conditions by several mechanisms. Also, authors have shown studies demonstrating beneficial effects of mitochondrial transfer in the brain under those neuropathologies. However, it would be better understood if authors could add summarized tables from findings to clearly illustrate any of the contents, such as models, experimental protocols, major findings, and interpretations among those reports. 

Author Response

In the present review article, the authors have well described the connection between neurons and glia during pathological conditions by several mechanisms. Also, authors have shown studies demonstrating beneficial effects of mitochondrial transfer in the brain under those neuropathologies. However, it would be better understood if authors could add summarized tables from findings to clearly illustrate any of the contents, such as models, experimental protocols, major findings, and interpretations among those reports. 

Answer: Two tables have been added to the manuscript. Table 1 summarizes the mechanisms of transfer used by cells in the brain (EVs or TNTs). Table 2 summarizes the various brain diseases in which mitochondria transfer has been investigated, and it’s therapeutic effects.

Reviewer 3 Report

A very interesting and well-written review article. The amount of analytical work done is impressive. I have only a few wishes for the authors.

1) The abstract should be written in more detail so that the reader can see the entire volume of the analytical material presented

2) The limitations and side effects of using mitochondrial transplantation should be clearly stated

Author Response

A very interesting and well-written review article. The amount of analytical work done is impressive. I have only a few wishes for the authors.

1) The abstract should be written in more detail so that the reader can see the entire volume of the analytical material presented

Answer: Additional sentences have been added as follows:

Lines 13-14: “In particular, artificial mitochondria transfer has sparked widespread interest as a potential therapeutic strategy for brain disorders.”

2) The limitations and side effects of using mitochondrial transplantation should be clearly stated

Answer: The following side effects has been added to section 6.2:

Lines 586-589: “In addition, the use of stem cells for AMT may increase the risk of developing tumors, as secondary cancers have been identified as a late complication of stem cell transplantation in humans [113].”

Limitations and side effects are discussed as follows:

-Difficulty in crossing the blood brain barrier (line 537-538)

-Invasiveness of procedure and poor translational value (line 543-545)

-Non-specific effects and uneven cellular distribution (line 548-551)

-limited studies comparing efficacy of different methods (line 555-556)

-Limited studies on long term effects (578-579)

Round 2

Reviewer 1 Report

We appreciate the comments of the authors, and the revisions made. In general, the review is very good and the relevant bibliography in the field have been mentioned. We congratulate the authors for the good work performed in this review.

However, we think that some of the comments were not well understood by the authors. We understand that some of the points were discussed in depth in the review, but our suggestion was to mention them earlier in a brief manner (with the relevant references) to make easy to follow the review.

Minor comments:

Title: Although we didn’t ask specifically for that, the changes in the title of the review to Mitochondria Transfer in Brain Injury and Disease is good.

The end of the paragraph about the reduced ability of oligodendrocytes to internalize mitochondria does not connect with the rest of the paragraph.

Answer: We agree with the referee. A linking sentence has been added to better connect this sentence with the rest of the paragraph as follows:

lines 96-100 “Fewer studies have investigated mitochondrial transfer in oligodendrocytes in the brain, however, Zhao and colleagues [34] reported that oligodendrocytes exhibit reduced capacity to internalize exogenous mitochondria, with less than 10% of oligodendrocytes exhibiting internalized mitochondria. Thus, they are not discussed in detail in this review.”

ü  OK, now the text is logical.

Detailed comments:

State and fate of mitochondria: Only in line 86 they mention the transfer of disrupted mitochondria from neurons to astrocytes for disposal. They should also add the reference of Davis et al, 2014 10.1073/pnas.1404651111. And also more recent paper from Lampinen et al, 2022 "Neuron-astrocyte transmitophagy is altered in Alzheimer's disease"10.1016/j.nbd.2022.105753

Answer: We thank the referee for this comment. Both references have now been added. Please see lines 88.

Furthermore, the transfer of healthy mitochondria from astrocytes to neurons has been described as a neuroprotective mechanism in Hayakawa, et al 2016 10.1038/nature18928.

Answer: The paper by Hayakawa et al. 2016 is mentioned twice in this paragraph (lines 82 & 87).

This paper is also discussed in more detail in later sections, including lines 182-188, 314-317, 433-466.

ü  Correct

Transfer of mitochondria in Glioblastoma

It has been described the transfer of mitochondria in brain tumor between tumor-to-tumor cell and between tumor cell-to-tumor microenvironment cell.

Between tumor-to-tumor cells: In glioblastoma stem-like cells these this transfer of mitochondria has been correlated with different responds of the glioblastoma stem-like cells to irradiation treatment Pinto, et al 2020 10.1042/BCJ20200710.

Between tumor and tumor microenvironment:

Interchange between glioblastoma and non-tumoral astrocytes: which can have a astrocytic protection role under hypoxic conditions Valdebenito et al 2021 10.1038/s41598-021-93775-8. And the association on delivery non-neoplastic mitochondria on glioblastoma drug response as well as proliferation and migration Civita et al, 2019 10.3390/ijms20236017.

Answer: We thank the referee for making us aware of this point. Transfer of mitochondria in glioblastoma has now been discussed and the relevant references added in section 2, lines 93-96 as follows:

“Mitochondria transfer has also been observed between tumor-to-tumor cells, and tumor-to-tumor microenvironment in the brain. For instance, mitochondria transfer has been observed between glioblastoma stem-like cells in vitro [36], as well as between glioblastoma cells and surrounding non-tumor astrocytes [37].”

ü  Correct

Paragraph 3: "Structural Mechanisms of Mitochondrial Transfer"

I suggest adding a table summarizing the disease in which mitochondria transfer via TNTs or EVs has been described, the status of the mitochondria, the consequence of the transfer if known, and the respective references.

Answer: We agree with the referee. Two tables have been added to the manuscript. Table 1 summarizes the mechanisms of transfer used by cells in the brain (EVs or TNTs). Table 2 summarizes the various brain diseases in which mitochondria transfer has been investigated, and its therapeutic effects.

Table 1. Mechanisms of Mitochondria Transfer in the Brain.

ü  Thanks, this table is very useful.

Table 2. In Vivo Artificial Mitochondria Transfer in Brain Injury and Disease

û  In Table 2, one of the columns is labeled "Delivery Method" and is confusing, they seem to refer to the method by which donor cells deliver mitochondria but, they mention the method the researchers used to study mitochondria transfer.

Extracellular Vesicles:

Again, the authors must specify under what conditions the EVs with mitochondria inside are delivered (stress conditions, diseases, etc.) and the consequences of the uptake of those EVs in the acceptor population.

Line 106. For instance, the small EVs Secreted by Nigrostriatal Astrocytes Rescue Cell Death and Preserve Mitochondrial Function in Parkinson's Disease (Leggio et al, 2022 doi.org/10.1002/adhm.202201203).

Answer: We thank the referee for this comment. A table has been added (table 1, line 168) summarizing the conditions in which EV-mediated mitochondria transfer occurs in the brain, and the consequences of their uptake.

The paper by Leggio et al. suggested by the reviewer was not added or discussed in the manuscript, because the EVs observed in this study did not contain mitochondria.

ü  OK

Tunneling nanotubes:

Major comments

This paragraph should be improved, the description of the Tunneling nanotubes is not accurate and the references are not the more relevant in this field. Furthermore, they should mention the transfer of mitochondria via TNTs in different disease and the consequences, for example: the transfer of mitochondria via TNTs have been described to rescues apoptotic PC12 (Wang et Gerdes, 2015 https://doi.org/10.1038/cdd.2014.211)

Answer: We thank the referee for this comment. A table has been added (table 1, line 168) summarizing the conditions in which TNT-mediated mitochondria transfer occurs in the brain, and the consequences of their uptake.

The paragraph has been edited to include the suggestions of the reviewer as outlined in the next two comments.

The paper by Wang and Gerdes, 2015 is referenced in the table.

Comment: the idea of adding this reference was to highlight the importance that the transfer of mitochondria via TNTs can have, more than just put only the reference in the table.

Detailed comments:

- They should mention the first time that this type of communication was described in by the group of Gerdes (Rustom et al, 2004. 10.1126/science.1093133).

Answer: We agree. The sections have been updated lines 120-122 as follows: “Tunneling nanotubes (TNTs) were first described in 2004 by Rustom and colleagues [42], who identified membranous channels comprised of F-actin that connect two or more cells and are capable of transferring mitochondria from one cell to another [43].”

ü  Correct

The definition "Tunneling nanotubes (TNTs) are membranous channels comprised of F-actin that connect two or more cells and transfer cytoplasmic molecules from one cell to another" is not complete because TNTs don’t only transfer cytoplasmatic molecules, they are able to transfer viruses, bacteria, and prions which use them as a route of propagation of the disease (Pepe A et al, 2022 10.1126/sciadv.abo0171, Onfelt et all 2006 10.4049/jimmunol.177.12.8476; Gousset et al. 2009 10.1038/ncb1841). Furthermore, they can transfer nanoparticles that could be use in the future as a nanocarrier mechanism of drug delivery (Saenz-de-Santa-Maria I et al, 2017 10.18632/oncotarget.15467). I would therefore specifically mention transfer of organelles such as mitochondria instead of "transfer cytoplasmic molecules" this is very ambiguous. This will be more coherent with the title of the section.

Answer: We agree with the Reviewer’s comments and have changed the section as follows:

“Tunneling nanotubes (TNTs) were first described in 2004 by Rustom and colleagues [42], who identified membranous channels comprised of F-actin that connect two or more cells and are capable of transferring mitochondria from one cell to another [43].”

Comment: a general sentence mentioning that tunneling nanotubes are used not only for organelle transfer is necessary. It has been described that TNTs can transfer α-synuclein, viruses, bacteria, and prions, which use them as a route of propagation of the disease.  It is therefore relevant to set up the importance of this type of communication particularly in the brain.

Other Mechanisms: As the authors say the transfer of mitochondria via gap junctions or cell fusions needs further investigations and it is not clear.

Answer: We thank the referee for this comment. This paragraph simply outlines that although other mechanisms of mitochondria transfer (gap junctions, cell fusion) have been identified in cell types outside of the brain, these methods of transfer have not yet been shown in brain cells. Thus, further investigation is required to elucidate whether these transfer mechanisms also occur in brain cells. We feel this point is clearly made, and so it has not been edited.

ü  OK

Figure 1: Intercellular mitochondrial transfer between neurons and astrocytes. Scheme of the different mechanism proposed for mitochondrial transfer between neurons and astrocytes. The cartoon of the figure is confusing. It looks like the mitochondria transfer occurs only via EVs or gap junctions. The authors should draw a TNT connecting the astrocyte and the neuron.

As an example of TNTs mediated transfer of mitochondria, the authors made a magnification of a single yellow mitochondrion (astrocytic mitochondria) that is hovering around the cells and draw the microtubules motor proteins that could be implicated in the transfer of mitochondrion via TNTs. This is incomplete because the motor proteins implicated in the transfer of mitochondria via TNTs are yet unknown.

In my opinion they should add the possibility of transfer neuronal mitochondrion (blue mitochondrion) from the neuron to the astrocytes. In that case, the proposed motor proteins (miro1/2, Milton and Kinesin), cannot be implicated in the transfer of mitochondria through TNTs formed by the neurons because they only contain actin filaments, as shown in the case of neuronal-like cells such as CAD cells and human neuroblastoma SH-SY5Y cells have TNTs only actin positive without microtubules (Sartori-Rupp et al, 2019 10.1038/s41467-018-08178-7). Therefore, they should add other candidates such as actin driven motor proteins: This should be also described in the text, with the corresponding references.

Answer: We thank the reviewer for making us aware of this point. We have realized that there were discrepancies between the figures in the word file and the pdf (probably due to issues with the file conversion for .doc to .pdf). Therefore, some items present in the figure 1 disappeared, namely the TNTs. The problem is fixed now, and the image was modified according to the reviewer comment. We apologize for the circumstances.

ü  Now the figure is improved, and very clear to follow, we congratulate the authors.

Regarding the motor proteins implicated in the transfer of mitochondria via TNTs, the Miro-1 protein was clearly identified in a study focused on mitochondrial transfer between neurons and astrocytes (ref. 97). Indeed, they showed that knocking down Miro-1 in astrocytes reduced mitochondrial transfer from astrocytes to neurons. This study was mentioned in the text lines 468-480.

Comment: We think that the authors misunderstood our comment. The point was to speculate that this transfer via Miro-1 is microtubule-dependent so, neuronal cells do not have microtubule-positive TNTs, so the TNT formed between astrocytes and neurons using Miro-1 could be a TNT formed from astrocytes and contain microtubules

 We also mention the study from Sartori-Rupp and colleagues which states that TNTs between neuronal cells are only composed of F-actin. Please see lines 138-141:

“Of note, Sartori-Rupp and colleagues found that mitochondria can be transported be-tween neuronal cells via individual TNTs composed of a single continuous bundle of parallel actin filaments [48]. This suggest that mitochondria directly bind to actin filaments and are transported via a microtubule- independent mechanism that remain to identify.”

ü  OK

Paragraph 4: "Effects of Mitochondrial Transfer in the Brain "

4.1. Enhancement of Cell Viability

The authors named the correct references, but they should discuss more in depth the work of Hayakawa et al, 2016 Transfer of mitochondria from astrocytes to neurons after stroke https://doi.org/10.1038/nature18928 and the work of English et al, 2020 as astrocytes transfer healthy mitochondria to neurons after cisplatin treatment (https://doi.org/10.1186/s40478-020-00897-7 ).

Answer: These papers are discussed in greater detail in the section “Mitochondria Transfer in Brain Injury and Disease” as follows:

Lines 314-317: “Hayakawa and colleagues showed that an influx of astrocytic mitochondria into adjacent neurons is observed following induction of focal cerebral ischaemia in mice, and this process has been linked to increased cell survival signals in neurons [20].”

And

Lines 409-412: “For instance, mitochondrial transfer from astrocytes increased neuronal survival, restored neuronal mitochondrial membrane potential, and normalized neuronal calcium dynamics in neurons treated with the chemotherapeutic drug cisplatin [18].”

transported via a microtubule- independent mechanism that remain to identify.”

ü  OK

4.2. Enhancement of Mitochondrial Degradation

Another example that they don’t mention is the important work of Sharma et al, 2019 that described how Rhes protein is able to travel via TNTs between striatal neuronal cell line from a healthy cell to the neighboring cells.

Answer: This paper has now been discussed lines 208-212 as follows:

“In addition, Rhes protein, a critical regulator of mitophagy in the brain, has been shown to transfer between striatal neuronal cells via TNTs, where it binds to damaged mitochondria in the recipient cell, suggesting that neurons may also transfer mitophagy-enhancing proteins to aid the transmitophagy process [57].”

ü  Correct

Figure2: Beneficial effects of intercellular mitochondria transfer in the brain.

The idea of the figure is good, but they should add the name of the cells next to the drawing- astrocytes next to the yellow cell, microglia the violet cells, and neuron blue cell.

Answer: We thank the reviewer for making us aware of this point. As stated for the Figure 1, we have realized that there were discrepancies between the figures in the word file and the pdf (probably due to issues with the file conversion for .doc to .pdf). The problem is fixed now.

ü  Correct

4.4. Deleterious Effects

They should add some reference about the mitochondria hijacked in neuronal cell under infection with SARS-CoV-2.

Answer: This reference has been added in the following section, lines 289-292:

“These findings also have implications for other disease-related cargo that are known to bind to mitochondria, such as SARS-CoV-2 [67], and suggest that under specific disease conditions, mitochondria may be hijacked as a means to facilitate cell-to-cell transfer of pathogens in the brain.”

ü  Correct

Paragraph 5: "Mitochondrial Transfer in Brain Injury and Disease "

5.4 Chemotherapy

They should add also in this section the effect of radiotherapy, and cite/discuss the work of Pinto et al, 2020 https://doi.org/10.1042/BCJ20200710

Answer: This paper has now been discussed lines 414-417 as follows:

“In addition, irradiation has been shown to have differing effects on TNT induction and mitochondria transfer in glioblastoma stem-like cells in vitro, depending on the time-course of treatment [36].”

ü  Correct

Paragraph 6: " Therapeutic Strategies Targeting Mitochondrial Transfer "

A missing possible therapeutic strategy directed to induce the mitophagy of disrupted mitochondria in astrocytes as a mechanism of clearance.

Furthermore, I would add a paragraph about therapeutic strategics to impair the transfer of healthy mitochondria in cancer as a mechanism of rescue or evasion of the apoptosis.

Answer: The therapeutic potential of impairing mitochondria transfer to cancerous cells is now mentioned as follows:

Lines 417-419: “These findings have important implications for whether mitochondrial transfer should be used as therapeutic treatment in cancer, as it may be used by tumor cells as a rescue mechanism, for evasion of apoptosis and tumor progression.”

Lines 429-430: “Of note, this research may also inform strategies to impair the transfer of healthy mitochondria to cancerous cells during chemotherapy.”

ü  OK

Paragraph 7: "Conclusions and Future Directions "

The authors develop the idea of the neuroprotective aspect of mitochondria transfer from glial cells to neurons and mitochondrial transplantation as a treatment for neurodegenerative diseases and brain injury. However, they have not considered the degradation of dysfunctional mitochondria by astrocytes which is key as a neuroprotective mechanism, and the development of therapeutic strategies to increase mitophagy of dysfunctional mitochondria. Most importantly, it is not considered how in cancer mitochondrial transfer can be used by tumor cells as a rescue mechanism, evasion of apoptosis and tumor progression. I believe that these ideas should be developed in this section of the review.

Answer: The degradation of dysfunctional mitochondria is mentioned in line 620.

The effects of mitochondria transfer in cancer is now addressed as follows:

Lines 416-419: “These findings have important implications for whether mitochondrial transfer should be used as therapeutic treatment in cancer, as it may be used by tumor cells as a rescue mechanism, evasion of apoptosis and tumor progression.”

ü  OK

Author Response

We thank the referee for all the comments. Please see below our answer. New edits are highlighted in green in the text.

  1. Paragraph 3: "Structural Mechanisms of Mitochondrial Transfer"

I suggest adding a table summarizing the disease in which mitochondria transfer via TNTs or EVs has been described, the status of the mitochondria, the consequence of the transfer if known, and the respective references.

Answer1: We agree with the referee. Two tables have been added to the manuscript. Table 1 summarizes the mechanisms of transfer used by cells in the brain (EVs or TNTs). Table 2 summarizes the various brain diseases in which mitochondria transfer has been investigated, and its therapeutic effects.

Table 1. Mechanisms of Mitochondria Transfer in the Brain.

ü  Thanks, this table is very useful.

Table 2. In Vivo Artificial Mitochondria Transfer in Brain Injury and Disease

û  In Table 2, one of the columns is labeled "Delivery Method" and is confusing, they seem to refer to the method by which donor cells deliver mitochondria but, they mention the method the researchers used to study mitochondria transfer.

Answer2: We thank the referee for making us aware of this point. To avoid any confusion, and emphasize that we refer only to artificial mitochondria transplantation, we changed the title of the table 2 to “Table 2. In Vivo Artificial Mitochondria Transplantation in Brain Injury and Disease”. We also precise in the column: “Method of Delivery for AMT (artificial mitochondria transplantation)”. Please see lines 611.

  1. Tunneling nanotubes:

Major comments

This paragraph should be improved, the description of the Tunneling nanotubes is not accurate and the references are not the more relevant in this field. Furthermore, they should mention the transfer of mitochondria via TNTs in different disease and the consequences, for example: the transfer of mitochondria via TNTs have been described to rescues apoptotic PC12 (Wang et Gerdes, 2015 https://doi.org/10.1038/cdd.2014.211)

Answer1: We thank the referee for this comment. A table has been added (table 1, line 168) summarizing the conditions in which TNT-mediated mitochondria transfer occurs in the brain, and the consequences of their uptake.

The paragraph has been edited to include the suggestions of the reviewer as outlined in the next two comments.

The paper by Wang and Gerdes, 2015 is referenced in the table.

Comment: the idea of adding this reference was to highlight the importance that the transfer of mitochondria via TNTs can have, more than just put only the reference in the table.

 Answer2: We thank the referee for this comment. We added the following sentence, lines 139-141:”Table 1 summarizes findings of studies investigating the effects of mitochondria transfer via TNTs in different brain disease models, and highlight the beneficial effect of this transfer [19,33,36,44,45,48-50].”

  1. The definition "Tunneling nanotubes (TNTs) are membranous channels comprised of F-actin that connect two or more cells and transfer cytoplasmic molecules from one cell to another" is not complete because TNTs don’t only transfer cytoplasmatic molecules, they are able to transfer viruses, bacteria, and prions which use them as a route of propagation of the disease (Pepe A et al, 2022 10.1126/sciadv.abo0171, Onfeltet all 2006 10.4049/jimmunol.177.12.8476; Gousset et al. 2009 10.1038/ncb1841). Furthermore, they can transfer nanoparticles that could be use in the future as a nanocarrier mechanism of drug delivery (Saenz-de-Santa-Maria I et al, 2017 10.18632/oncotarget.15467). I would therefore specifically mention transfer of organelles such as mitochondria instead of "transfer cytoplasmic molecules" this is very ambiguous. This will be more coherent with the title of the section.

Answer1: We agree with the Reviewer’s comments and have changed the section as follows:

“Tunneling nanotubes (TNTs) were first described in 2004 by Rustom and colleagues [42], who identified membranous channels comprised of F-actin that connect two or more cells and are capable of transferring mitochondria from one cell to another [43].”

Comment: a general sentence mentioning that tunneling nanotubes are used not only for organelle transfer is necessary. It has been described that TNTs can transfer α-synuclein, viruses, bacteria, and prions, which use them as a route of propagation of the disease.  It is therefore relevant to set up the importance of this type of communication particularly in the brain.

Answer2: We thank the referee for this comment. We revised the definition accordingly. Please see lines 120-124: “Tunneling nanotubes (TNTs) were first described in 2004 by Rustom and colleagues [42], who identified membranous channels comprised of F-actin that connect two or more cells and are involved in cell-to-cell communication [43]. TNTs were shown to transfer different organelles, including mitochondria, from one cell to another, but also other cargos including proteins (e.g. α-synuclein) and nucleotides [42-44].”